# Agent Data Protocol: Unifying Datasets for Diverse, Effective Fine-tuning of LLM Agents

**Yueqi Song[1], Ketan Ramaneti[1], Zaid Sheikh[1], Ziru Chen[2], Boyu Gou[2], Tianbao Xie[3], Yiheng Xu[3], Danyang Zhang[3], Apurva Gandhi[1], Fan Yang[5], Joseph Liu[1], Tianyue Ou[1], Zhihao Yuan[1], Frank Xu[1], Shuyan Zhou[4], Xingyao Wang[6], Xiang Yue[1], Tao Yu[3], Huan Sun[2], Yu Su[2], Graham Neubig[1,6]**

[1]Carnegie Mellon University, [2]The Ohio State University, [3]University of Hong Kong,
[4]Duke University, [5]Fujitsu Research, [6]All Hands AI
{yueqis,gneubig}@cs.cmu.edu

⬡ https://agentdataprotocol.com

## ABSTRACT

Public research results on large-scale supervised finetuning of AI agents remain relatively rare, since the collection of agent training data presents unique challenges. In this work, we argue that the bottleneck is not a lack of underlying data sources, but that a large variety of data is fragmented across heterogeneous formats, tools, and interfaces. To this end, we introduce the Agent Data Protocol (ADP), a light-weight representation language that serves as an "interlingua" between agent datasets in diverse formats and unified agent training pipelines downstream. The design of ADP is expressive enough to capture a large variety of tasks, including API/tool use, browsing, coding, software engineering, and general agentic workflows, while remaining simple to parse and train on without engineering at a per-dataset level. In experiments, we unified a broad collection of 13 existing agent training datasets into ADP format, and converted the standardized ADP data into training-ready formats for multiple agent frameworks. We performed supervised finetuning on the unified data, and demonstrated an average performance gain of ~20% over corresponding base models, and delivers state-of-the-art or near-SOTA performance on standard coding, browsing, tool use, and research benchmarks, without domain-specific tuning. All code and data are released publicly, in the hope that ADP could help lower the barrier to standardized, scalable, and reproducible agent training.

## 1 INTRODUCTION

Pre-training large language models (LLMs) benefits from abundant, readily available Internet-scale data. In contrast, post-training presents a much harder challenge: high-quality task-specific data must be carefully curated. While creative strategies have emerged for collecting data in relatively simple settings, such as single-turn user interactions like code generation (Nijkamp et al., 2023), question answering (Rajpurkar et al., 2016), and sentiment analysis (Maas et al., 2011), many real-world tasks are far more complex.

A particularly difficult case is agent applications, where models must take sequential actions and interact with the world iteratively. Building datasets for such scenarios requires recording and structuring trajectories of agent behavior, much more challenging than collecting static input-output pairs.

Despite these difficulties, a growing body of work has explored different approaches for creating agent datasets. These efforts vary in methodology, from manual curation (Rawles et al., 2023; Xu et al., 2024a), to synthetic data generation (Ou et al., 2024; Zheng et al., 2024a), to recorded agent rollouts (Pan et al., 2025; Yang et al., 2025b). The resulting datasets span a wide range of tasks, including web navigation (Deng et al., 2023; Lù et al., 2024), software development (Yang et al., 2025b; Pan et al., 2025), visual interface control (Rawles et al., 2023; Kapoor et al., 2024), and general tool use (Zeng et al., 2023; Liu et al., 2024a) (an overview of these datasets in § 2.1).

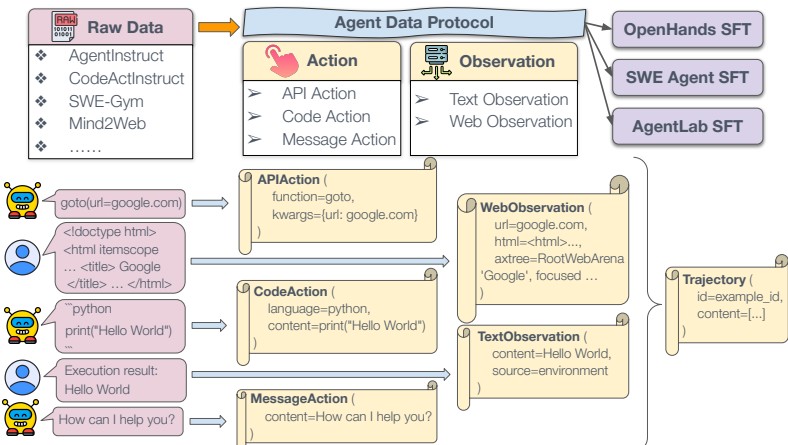

Figure 1: Overview of the Agent Data Protocol (ADP). Raw data from diverse sources such as SWE-Gym are converted into a standardized ADP format. ADP unifies data into Trajectory objects, which include two core components: Actions (API action, code action, message action) and Observations (text observation, web observation), enabling seamless integration with various agent SFT formats. Example conversions show how heterogeneous raw data is normalized for training agentic models.

However, despite the availability of such data, large-scale supervised fine-tuning (SFT) of agents remains rare in academic research. A few notable projects, such as Zeng et al. (2023) and Mitra et al. (2024), have demonstrated their potential, but remain exceptions rather than the norm. Why has this not become standard practice? We argue that *the issue is not a lack of data, but rather a lack of standardization*. Existing datasets are fragmented, with inconsistent formats and representations, making it difficult to combine, share, and leverage them effectively, thus they remain underutilized.

To address this gap, we introduce the Agent Data Protocol (ADP), a standardized expressive representation language for agent data. By converting heterogeneous datasets into ADP, it makes it simple to generate large-scale and diverse data for a variety of downstream training pipelines (Figure 1). Technically, ADP is implemented as Pydantic[1] schemas that express actions and observations corresponding to common agent use cases such as communicating, browsing, coding, and miscellaneous tool calling, coupled with strict automated validation to maintain high data quality.

As a first step to demonstrate the practical utility of ADP, we implement converters from 13 pre-existing datasets into ADP, and converters from ADP to 3 different agent architectures, demonstrating its generality. Based on this, we create and release the largest publicly available dataset for agent training, consisting of 1.3M training trajectories, dubbed the ADP Dataset V1.

Our experiments show training agents using ADP leads to significant performance improvements across diverse domains, including coding (SWE-Bench Verified), web browsing (WebArena), research (GAIA), and agentic tool use (AgentBench), as shown in § 6. Notably, these results improve by an average of 20% over base models, and are competitive with or superior to other state-of-the-art results from similarly-sized models. We also identify significant benefits from cross-task transfer, with training on the ADP data improving significantly over training on individual datasets. Beyond performance, ADP enables systematic cross-dataset analysis, revealing trends and areas for improvement in publicly available data.

Finally, we release all code and datasets in open source to foster community adoption and encourage contributions of new datasets. We believe ADP will unlock a new wave of progress in agentic model fine-tuning by providing the standardization needed to make large-scale supervised agent training practical and scalable.

## 2   RELATED WORK

The development of effective LLM-based agents critically depends on high-quality training data that could capture the complexity of multi-step reasoning, tool usage, and environmental interaction

---

[1] https://pydantic.dev/

(Yao et al., 2022b; Schick et al., 2023; Deng et al., 2023; Masterman et al., 2024). This section reviews existing methods for agent data collection and the challenges that motivate ADP.

## 2.1 Agent Data Collection Methods

Existing approaches span manual creation (human experts creating step-by-step demonstrations of desired agent behaviors) (Nakano et al., 2021; Yao et al., 2022a), synthetic generation (leverages existing LLMs to create agent trajectories through prompting or structured generation) (Luo et al., 2023; Xu et al., 2024b), and recorded agent rollouts (captures trajectories from existing agent systems during task execution) (Wang et al., 2024a; Pan et al., 2025), etc, resulting in abundant agent training data, a representative set of which listed in Table 1.

Table 1: Overview of Existing Agent Training Datasets. C=Coding, S=Software Engineering, T=API/Tool Use, W=Web Browsing.

| Dataset | Variety | Count | Source | Note |
|---|---|---|---|---|
| **AgentInstruct** (Zeng et al., 2023) | C T W | 1.9K | synthetic | Mixture of Browsing, Database, OS, etc. |
| **Code-Feedback** (Zheng et al., 2024a) | C | 66.4K | manual | Code generation with runtime feedback loops |
| **CodeActInstruct** (Wang et al., 2024b) | C | 7.1K | synthetic | Code generation and tool use with execution |
| **Go-Browse**(Gandhi & Neubig, 2025) | W | 9.5K | rollout | Structured exploration web rollouts |
| **Mind2Web** (Deng et al., 2023) | W | 1.0K | manual | Human web demos on real websites |
| **Nebius SWE Trajectories** (Golubev et al., 2024) | S | 13.4K | rollout | SWE-agent trajectories from Nebius relying solely on open-weight models |
| **NNetNav-live** (Murty et al., 2024) | W | 5.0K | rollout | Retroactively labeled live web exploration |
| **NNetNav-wa** (Murty et al., 2024) | W | 4.2K | rollout | Retroactively labeled WebArena exploration |
| **openhands-feedback** (All Hands AI, 2024) | C T W | 0.2K | rollout | Recorded OpenHands agent trajectories with human feedback |
| **Orca Agentinstruct** (Mitra et al., 2024) | T | 1046.1K | synthetic | Large-scale synthetic tool-use instructions data |
| **SWE-Gym** (Pan et al., 2025) | S | 0.5K | rollout | Agent trajectories solving real GitHub repo tasks |
| **SWE-smith** (Yang et al., 2025b) | S | 5.0K | synthetic | Trajectories of agents on synthesized bug-fix tasks |
| **Synatra** (Ou et al., 2024) | W | 99.9K | synthetic | Synthetically created web demos of tutorials |

We also group each dataset into a coarse task category.

- **Coding**: generally includes fundamental programming tasks, such as command line code generation, algorithm implementation, code completion, code translation, and code repair, etc.

- **Software Engineering**: often consists of repository-level software engineering tasks, such as bug fixing, feature implementation, code refactoring, and dependency management, etc.

- **API/Tool Use**: usually requires agents to use external APIs/tools effectively to solve tasks. Common tools include file manipulation, database queries, and customized APIs, etc.

- **Web Browsing**: commonly encompasses tasks including web navigation, online shopping, and social media interactions, etc, requiring agents to understand GUIs.

## 2.2 Challenges and Limitations

Despite abundant existing agent training datasets, several fundamental challenges prevent effective large-scale utilization of these resources:

- **Complexity of Data Curation**: Creation of high-quality agent training data requires significant resources and expertise (Paullada et al., 2021; Bhardwaj et al., 2024; Zha et al., 2025). Manual curation is expensive and requires domain knowledge; synthetic generation faces challenges in verifying data quality; recorded agent rollouts are fundamentally constrained by the capabilities of existing baseline agents, limiting the diversity and complexity of trajectories. While recent efforts have scaled trajectory collection (Song et al., 2024; Mitra et al., 2024), the fundamental challenge of balancing quality, diversity, and scale across different curation approaches remains.

- **Heterogeneity of Dataset Format**: Existing agent training datasets each employ its own representation format, action spaces, and observation structures (Ning et al., 2025; Luo et al., 2025). For example, some web datasets use HTML while some use accessibility tree structures (de Chezelles et al., 2025). Existing efforts have noted and begun addressing data standardization (Zhang et al., 2024; Chen et al., 2024; Mohammadi et al., 2025; Xi et al., 2025; Zhang et al., 2025), but they mostly focused on proposing task-specific or agent-specific unification rather than community-wide standardization of data representation, limiting plug-and-play with other datasets or agents,

where significant engineering effort is still required to utilize multiple datasets together, hindering integration across different data sources.

- **Difficulty of Analysis and Comparison**: The diverse structures of existing datasets also makes it difficult to perform systematic comparisons or quantitative analysis across different data sources (Putrama & Martinek, 2024), limiting researchers' ability to understand the relative usefulness, coverage, and quality of different datasets, hindering data-driven selection or improvements.

## 3 THE AGENT DATA PROTOCOL

To overcome these challenges and limitations, and to make good use of existing data resources, we propose the Agent Data Protocol (ADP). ADP establishes a unified schema that bridges the gap between existing heterogeneous agent training datasets and large-scale supervised agent fine-tuning.

### 3.1 DESIGN PRINCIPLES

We design ADP around the following core principles:

- **Simplicity**: ADP maintains a simple and intuitive structure. This directly addresses the *complexity of data curation* challenge by providing a straightforward framework that eliminates the need for specialized per-dataset engineering, making large-scale agent data utilization accessible to researchers without extensive adaptation effort.

- **Standardization**: ADP is designed to provide a unified representation that unifies existing agent training datasets of various different formats to a standardized format, addressing the challenge of *heterogeneous dataset formats*.

- **Expressiveness**: ADP is designed to ensure that complex agentic trajectories could be accurately expressed with no loss of critical information. This directly addresses the *difficulty of analysis and comparison* challenge because ADP is expressive enough to cover the broad variety of existing agent datasets across different domains, enabling researchers to put these diverse datasets under the same conditions and context.

By addressing the fundamental challenges in utilization agent data, ADP aims to push the progress in agent training, making large-scale agent SFT more accessible to the broader research community.

### 3.2 ARCHITECTURE

The ADP schema is implemented as Pydantic schemas, and is simple yet expressive in design. Each ADP standardized agent trajectory is represented as a `Trajectory` object.

**Trajectory** consists of (1) `id`: trajectory id, (2) `content`: an alternating sequence of actions and observations representing the agent's interaction with the user/environment, (3) `details`: A flexible metadata dictionary for dataset-specific information (e.g., dataset source URLs).

**Action** represents agents' decisions and behaviors. We categorize actions into three types:

- **API Actions**: Function calls with structured parameters and outputs capturing tool use. Each API action includes: (1) `function`: name of tool call, (2) `kwargs`: a dictionary of function arguments, and (3) `description`: optional reasoning or explanation for the action. For example, with ADP, a web navigation call `goto(url=https://www.google.com)` is represented as `APIAction(function=goto, kwargs=url:https://www.google.com)`.

- **Code Actions**: Code generation and execution across programming languages. Each code action specifies: (1) `language`: the programming language (e.g., python), (2) `content`: the code to execute, and (3) `description`: optional reasoning or explanation for the action. For example, the ADP representation of a python code block ` ```python print("Hello World")``` ` is `CodeAction(language=python,content=print("Hello World")`.

- **Message Actions**: Natural language communications between agents and users, each containing a `content` field, documenting agents' explanations, clarifications, and responses. For example, `MessageAction(content=How can I help you?)`.

**Observation** represents agents' perceptions from the environment, categorized into two types:

- **Text Observations**: Captures the text information from various sources, including user instructions and environmental feedback. Each text observation includes: (1) `source`: the origin of the observation ("user" or "environment"), and (2) `content`: the observed text. For example, a python execution output `Execution result:  Hello World`, will be converted to ADP format `TextObservation(content=Hellow World, source=environment)`.
- **Web Observations**: Represent the state and content of webpages. Each observation includes: (1) `html`: raw HTML content, (2) `axtree`: accessibility tree of the webpage, (3) `url`: current page URL, (4) `viewport_size`: browser viewport dimensions, and (5) `image_observation`: optional screenshot data. Web observations enable ADP to support complex browsing scenarios.

The core insight behind ADP is that despite the surface-level diversity in agent datasets, most agentic interactions can be decomposed into a sequence of *actions* taken by the agent and *observations* received from the environment. By standardizing these fundamental components, ADP directly addresses each challenge identified in § 2.2 while preserving the rich semantics of the original data. This unified representation enables researchers to combine datasets that were previously incompatible, facilitating large-scale training across diverse domains.

## 3.3 CONVERSION PIPELINE

As shown in Figure 1, we implemented a three-stage conversion pipeline with ADP that transforms heterogeneous datasets into training-ready agentic formats.

1. **Raw to Standardized**: This stage unifies original dataset formats into the ADP standardized schema. Each dataset is extracted in its raw format, and then converted to the ADP schema by mapping each dataset-specific actions and observations to the ADP's standardized action and observation space. For example, a web browsing task with HTML representations is converted to a pairs of `APIAction` and `WebObservation`, while a coding task with execution output is mapped to `CodeAction` and `TextObservation` pairs.

2. **Standardized to SFT**: This stage converts ADP standardized trajectories into supervised fine-tuning (SFT) format suitable for training language models. Different agent frameworks operate with distinct actions spaces, observations formats, etc. For example, OpenHands employs IPython execution with web browsing capabilities, SWE-Agent uses structured bash commands and file operations, while AgentLab focuses on DOM-based web interactions. Rather than training only one generic action model, we recognize that effective agent training requires adaptation to each framework's specific scaffolding and interactions formats. For each agent harness, the conversion process uses one agent-specific script that translates each type of action and observation into the target agent's action and observation space based on the agent's framework. This stage handles context management, specifies system prompts, and formats conversations to create SFT-ready instruction-response pairs, optimized for the particular agent architecture.

3. **Quality Assurance**: This stage ensures data correctness and consistency in alignment with agent format, tool use, and conversation structure through automated validation. Example quality checks include verifying tool call formats, ensuring most[2] tool calls are paired with an English thought, and checking whether the conversation ends properly, etc.

## 3.4 PRACTICAL IMPACT OF ADP ON AGENT TRAINING RESEARCH

The two-direction pipeline (Raw→ADP and ADP→SFT) cleanly separates responsibilities and eliminates redundant engineering (Figure 2). In practice:

- **Dataset conversion (once per dataset).** Contributors convert each *raw* dataset to the *ADP* schema *exactly once*. From then on, the dataset is a standardized resource usable by any agent harness.
- **Agent-specific conversion (once per agent).** Each agent maintains a single script for *ADP→SFT*; no per-dataset engineering needed. Adding new datasets requires *no* change to agent-side scripts.

---

[2]We set this threshold to be 80%, but it can be changed based on demand.

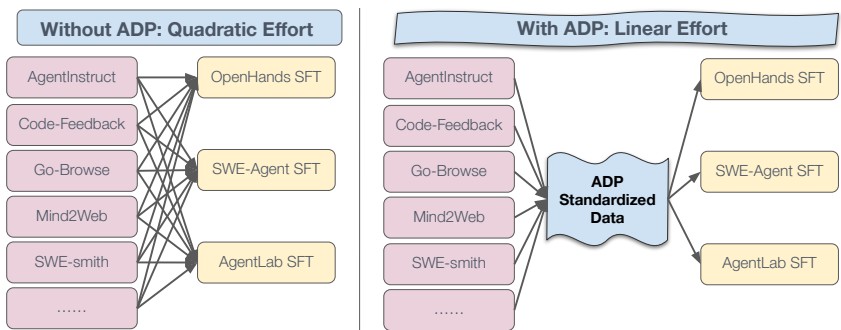

Figure 2: **ADP collapses many-to-many conversions into a hub-and-spoke pipeline.** *Left:* Without ADP, each of $D$-many datasets needs a custom Raw→SFT converter for each of $A$-many agentic formats (quadratic $O(D \times A)$ effort), causing duplicated code and efforts. *Right:* With ADP, each dataset is converted once (Raw→ADP) and each agent only requires one converter (ADP→SFT), yielding linear $O(D+A)$ effort. New datasets or agents plug in immediately to the rest of ADP.

ADP amortizes conversion cost across the community, accelerates adoption of new datasets, and ensures that a single ADP→SFT script instantly unlocks the entire pool of ADP-standardized data to an agent framework. Without ADP, researchers must write a *Raw→SFT* converter for *each* dataset–agent pair, duplicating effort across groups and making large-scale data integration brittle and slow. More discussion could be found in § 6.3.

## 4 CROSS DATASET ANALYSIS

Table 2 shows analysis on 13 ADP standardized datasets, revealing significant diversity across datasets.

**Trajectory Length.** Trajectory rounds vary dramatically across datasets, from 1 to 26.8 turns, with an average of 10.1 turns. SWE datasets consistently exhibit longer length, reflecting the inherent complexity of repo-level programming tasks.

**Action Distribution.** Clear domain-specific preferences emerge from the action distributions after standardization with ADP. Web datasets (e.g. Mind2Web) heavily favor API actions with minimal code execution, reflecting their focus on interface interaction. Conversely, coding datasets (e.g., CodeActInstruct) show high code usage with no API usage, emphasizing direct programming activities. SWE datasets (e.g., SWE-smith) demonstrate mixed patterns, which relies on API actions like file writes while using code actions for code generation.

Table 2: Dataset Stats and Trajectory Analysis. A=APIAction, C=CodeAction, M=MessageAction.

| Dataset | AVG. Rounds | % Actions (A/C/M) | % Func Thought |
|---|---|---|---|
| **AgentInstruct** | 8.2 | 64/10/26 | 100.0 |
| **Code-Feedback** | 4.0 | 0/58/42 | 82.8 |
| **CodeActInstruct** | 4.0 | 0/65/35 | 98.6 |
| **Go-Browse** | 3.9 | 70/0/30 | 100.0 |
| **Mind2Web** | 9.7 | 90/0/10 | 0.0 |
| **Nebius SWE-Agent** | 16.2 | 67/27/6 | 100.0 |
| **NNetNav-live** | 8.2 | 80/0/20 | 99.9 |
| **NNetNav-wa** | 10.1 | 89/0/11 | 99.9 |
| **OpenHands** | 18.3 | 11/73/16 | 91.7 |
| **Orca AgentInstruct** | 1.3 | 0/15/85 | 84.0 |
| **SWE-Gym** | 19.7 | 61/25/14 | 42.0 |
| **SWE-smith** | 26.8 | 56/40/4 | 90.1 |
| **Synatra** | 1.0 | 100/0/0 | 99.9 |
| **Overall** | **10.1** | **53/24/23** | **83.8** |

**Function Reasoning Analysis.** A striking finding is the high function thought coverage ($\geq 90\%$ for most datasets), indicating that these training datasets consistently provide explanations for actions. This is particularly valuable for interpretability and training agents with reasoning abilities. Importantly, high reasoning coverage appears across all task varieties, suggesting that function thoughts represent a general characteristic of well-documented datasets rather than domain-specific behavior.

## 5 EXPERIMENTAL SETUP

### 5.1 TRAINING SETUP

To evaluate ADP's effectiveness in training across diverse data sources, we utilize a comprehensive collection of 13 agent training datasets, spanning coding, SWE, API/tool user, and browsing, as

documented in Table 1. These datasets represent a broad spectrum of heterogeneity challenges that ADP addresses, including varied data creation methodologies (synthetic generation, manual curation, agent rollouts), different complexity (from simple to complex multi-step workflows), and diverse environments (command-line interfaces, web GUIs, Jupyter Notebooks, API calls).

The selected datasets collectively contain over 1.3M instances, ranging from smaller ones like Mind2Web to larger-scale ones like Orca AgentInstruct. To ensure balanced representation across domains and prevent any single large dataset from dominating the training process, we subsample from larger datasets while using smaller datasets in their entirety. Full details of our data sampling and mixture weights are in Appendix C.

We use Qwen2.5-Coder-Instruct model family (Qwen Team, 2024; Hui et al., 2024) as the base models, with 3 agent frameworks for comprehensive evaluation across multiple benchmarks. We fine-tuned all models using the same SFT pipeline from LLaMA-Factory (Zheng et al., 2024b). These experiments focus on each framework's specialized domain to demonstrate targeted effectiveness. Each agent has unique architectures, tool interfaces, and interaction environments. This diversity allows us to validate that ADP-standardized data can be readily and easily converted to different agent formats, demonstrating the protocol's utility across various agent implementations.

**OpenHands** (Wang et al., 2025) is an open platform for building generalist AI agents that operate like software developers: writing code, using command lines, and browsing the web. It provides sandboxed execution environments, tool coordination, and benchmark evaluation.

**AgentLab** (Drouin et al., 2024; de Chezelles et al., 2025) is an open-source framework for developing, testing, and benchmarking web agents across diverse tasks, emphasizing scalability and reproducibility. It supports a suite of evaluation benchmarks like WebArena and WorkArena.

**SWE-Agent** (Yang et al., 2024) introduces a custom Agent-Computer Interface (ACI) that enables language model agents to autonomously perform software engineering tasks by navigating codebases, editing and running code, viewing files, and executing tests.

## 5.2 EVALUATION BENCHMARKS

We evaluated these agents across 4 benchmarks (based on the availability of benchmark evaluation code and specialization of agents) that span different domains. This comprehensive evaluation demonstrates ADP's expressiveness in preserving critical information across diverse tasks.

**SWE-Bench** (Jimenez et al., 2024) evaluates agents on real-world software engineering tasks. Given a Github codebase and a bug report, agents must generate patches that satisfy existing unit tests. We used the SWE-Bench Verified subset for evaluation (Chowdhury et al., 2024).

**WebArena** (Zhou et al., 2024) provides a realistic, self-hosted web environment composed of fully functional websites in domains like e-commerce, forums, and map navigation, requiring agents to interpret high-level natural language commands and perform concrete web interactions.

**AgentBench** (Liu et al., 2024b) evaluates agents across different environments, such as operating systems, databases, and web browsing. It emphasizes multi-turn reasoning, decision making, and adaptability across domains.

**GAIA** (Mialon et al., 2023) is a benchmark for general AI assistants featuring human-annotated tasks that combine reasoning, tool use, and multi-step problem solving, often with multimodal input. Tasks vary in difficulty by number of steps and required tools.

## 6 EXPERIMENTAL RESULTS

### 6.1 ADP DATA RESULTS IN HIGHLY EFFECTIVE AGENTS ACROSS DIVERSE TASKS

**ADP fine-tuning consistently improves performance across models, benchmarks, and agent harnesses.** As shown in Table 3, Table 4, and Table 5, training on standardized ADP data yields substantial gains across 7B, 14B, and 32B models on several popular evaluation benchmarks. On *SWE-Bench (Verified)*, ADP training delivers remarkable improvements: `Qwen-2.5-7B-Coder-Instruct` improves from 0.4% to 20.2% (+19.8%)

Table 3: Comparison of SOTA and our Best 7–8B ADP-trained agents' results across benchmarks. Shaded rows are our ADP-tuned models. Other rows are collected from previous works.

| Agent | Model | Training Data | Accuracy |
|---|---|---|---|
| *SWE-Bench (Verified) (Jimenez et al., 2024; Chowdhury et al., 2024)* | | | |
| SWE-Agent (Yang et al., 2024) | Qwen-2.5-7B-Coder-Instruct | – | 0.4% |
| | Qwen-2.5-7B-Coder-Instruct | SWE-smith (Yang et al., 2025b) | 15.2% (+14.8%) |
| | Claude 3 Opus (Anthropic Team) | – | 15.8% |
| | Qwen-2.5-7B-Coder-Instruct | ADP Data | 20.2% (+19.8%) |
| OpenHands CodeActAgent (Wang et al., 2025) | Qwen-2.5-7B-Coder-Instruct | – | 2.8% |
| | Qwen-2.5-7B-Coder-Instruct | SWE-Gym (Pan et al., 2025) | 10.6% (+7.8%) |
| | Qwen-2.5-7B-Coder-Instruct | ADP Data | 20.4% (+17.6%) |
| *WebArena (Zhou et al., 2024)* | | | |
| BrowserGym (de Chezelles et al., 2025) | Llama-3.1-8B | – | 1.0% |
| | Qwen-2.5-7B-Instruct | – | 8.3% |
| | Llama-3.1-8B | NNetNav (Murty et al., 2024) | 16.3% (+15.3%) |
| | Qwen-2.5-7B-Instruct | Go-Browse (Gandhi & Neubig, 2025) | 21.7% (+13.4%) |
| AgentLab (Drouin et al., 2024) (de Chezelles et al., 2025) | Qwen-2.5-7B-Coder-Instruct | – | 4.5% |
| | Qwen-2.5-7B-Coder-Instruct | ADP Data | 21.0% (+16.5%) |
| *AgentBench OS (Liu et al., 2024b)* | | | |
| AgentLM (Liu et al., 2024b) | Llama-2-chat-7B | – | 8.3% |
| | Llama-2-chat-7B | AgentInstruct (Zeng et al., 2023) | 17.4% (+9.1%) |
| OpenHands CodeActAgent (Wang et al., 2025) | Qwen-2.5-7B-Coder-Instruct | – | 3.5% |
| | Qwen-2.5-7B-Coder-Instruct | ADP Data | 27.1% (+23.6%) |
| *GAIA (Mialon et al., 2023)* | | | |
| OWL Agent (Hu et al., 2025) | Qwen-2.5-7B-Instruct | – | 4.8% |
| OpenHands CodeActAgent (Wang et al., 2025) | Qwen-2.5-7B-Instruct | – | 7.3% |
| | Qwen-2.5-7B-Instruct | ADP Data | 9.1% (+1.8%) |

Table 4: Comparison of SOTA and our Best 13–14B ADP-trained agents' results across benchmarks. Shaded rows are our ADP-tuned models. Other rows are collected from previous works.

| Agent | Model | Training Data | Accuracy |
|---|---|---|---|
| *SWE-Bench (Verified) (Jimenez et al., 2024; Chowdhury et al., 2024)* | | | |
| SWE-Agent (Yang et al., 2024) | Qwen-2.5-14B-Coder-Instruct | – | 2.0% |
| | Claude 3.5 Sonnet(Anthropic Team) | – | 33.6% |
| | Qwen-2.5-14B-Coder-Instruct | ADP Data | 34.4% (+32.4%) |
| OpenHands CodeActAgent (Wang et al., 2025) | Qwen-2.5-14B-Coder-Instruct | – | 5.8% |
| | Qwen-2.5-14B-Coder-Instruct | SWE-Gym (Pan et al., 2025) | 16.4% (+10.6%) |
| | Qwen-2.5-14B-Coder-Instruct | ADP Data | 30.6% (+24.8%) |
| *WebArena (Zhou et al., 2024)* | | | |
| AgentLab (Drouin et al., 2024) (de Chezelles et al., 2025) | Qwen-2.5-14B-Coder-Instruct | – | 5.5% |
| | Qwen-2.5-14B-Coder-Instruct | ADP Data | 22.2% (+16.7%) |
| *AgentBench OS (Liu et al., 2024b)* | | | |
| AgentLM (Liu et al., 2024b) | Llama-2-chat-13B | – | 9.0% |
| | Llama-2-chat-13B | AgentInstruct (Zeng et al., 2023) | 18.1% (+9.1%) |
| OpenHands CodeActAgent (Wang et al., 2025) | Qwen-2.5-14B-Coder-Instruct | – | 2.8% |
| | Qwen-2.5-14B-Coder-Instruct | ADP Data | 20.8% (+18.0%) |

with SWE-Agent and from 2.8% to 20.4% (+17.6%) with OpenHands. At 14B scale, Qwen-2.5-14B-Coder-Instruct achieves 34.4% (+32.4%) with SWE-Agent and 30.6% (+24.8%) with OpenHands. The 32B model reaches 40.3% (+38.1%) with SWE-Agent and 36.8% (+26.2%) with OpenHands, matching or exceeding Claude 3.5 Sonnet with SWE-Agent's 33.6% performance. On *WebArena*, ADP training shows consistent gains across model sizes: 7B achieves 21.0% (+16.5%), 14B reaches 22.2% (+16.7%), and 32B attains 22.9% (+12.0%). On *AgentBench OS*, the improvements are substantial: the 7B model improves from 3.5% to 27.1% (+23.6%), the 14B model improves from 2.8% to 20.8% (+18.0%), and 32B models from 27.8% to 34.7% (+6.9%). Finally, on *GAIA*, the 7B model improves from 7.3% to 9.1% (+1.8%).

These gains, spanning both coding and browsing settings, show that a unified, cross-domain ADP training corpus can deliver SOTA or near-SOTA performance without domain-specific tuning and is effective across models, action spaces, and agent harnesses. Figure 3 and Figure 4 also show clear

Table 5: Comparison of SOTA and our Best 32B ADP-trained agents' results across benchmarks. Shaded rows are our ADP-tuned models. Other rows are collected from previous works.

| Agent | Model | Training Data | Accuracy |
|---|---|---|---|
| *SWE-Bench (Verified) (Jimenez et al., 2024; Chowdhury et al., 2024)* | | | |
| SWE-Agent | `Qwen-2.5-32B-Coder-Instruct` | – | 2.2% |
| (Yang et al., 2024) | `Qwen-2.5-32B-Coder-Instruct` | SWE-smith (Yang et al., 2025b) | 40.2% (+38.0%) |
| | `Qwen-2.5-32B-Coder-Instruct` | ADP Data | 40.3% (+38.1%) |
| OpenHands CodeActAgent | `Qwen-2.5-32B-Coder-Instruct` | – | 10.6% |
| (Wang et al., 2025) | `Qwen-2.5-32B-Coder-Instruct` | SWE-Gym (Pan et al., 2025) | 20.6% (+10.0%) |
| | `Qwen-2.5-32B-Coder-Instruct` | ADP Data | 36.8% (+26.2%) |
| *WebArena (Zhou et al., 2024)* | | | |
| AgentLab (Drouin et al., 2024) | `Qwen-2.5-32B-Coder-Instruct` | – | 10.9% |
| (de Chezelles et al., 2025) | `Qwen-2.5-32B-Coder-Instruct` | ADP Data | 22.9% (+12.0%) |
| *AgentBench OS (Liu et al., 2024b)* | | | |
| AgentLM | `Llama-2-chat-70B` | – | 9.0% |
| (Liu et al., 2024b) | `Llama-2-chat-70B` | AgentInstruct (Zeng et al., 2023) | 21.5% (+12.5%) |
| OpenHands CodeActAgent | `Qwen-2.5-32B-Coder-Instruct` | – | 27.8% |
| (Wang et al., 2025) | `Qwen-2.5-32B-Coder-Instruct` | ADP Data | 34.7% (+6.9%) |

monotonic gains with model size and consistent boosts from ADP training across agents and tasks, with ADP-trained models outperforming their base counterparts at every scale.

## 6.2 DIVERSE DATA RESULTS IN CROSS-TASK TRANSFER

Table 6: Cross-task transfer with diverse vs. task-specific data. For each benchmark, we compare the same harness+model under task-specific "only" tuning and training on ADP corpus.

| Agent | Model | Training Data | Accuracy |
|---|---|---|---|
| *SWE-Bench (Verified) (Jimenez et al., 2024; Chowdhury et al., 2024)* | | | |
| OpenHands CodeActAgent | `Qwen-2.5-7B-Instruct` | SWE-smith Only | 1.0% |
| (Wang et al., 2025) | `Qwen-2.5-7B-Instruct` | ADP Data | 10.4% |
| | `Qwen-3-8B` | CodeActInstruct + Code-Feedback | 0.2% |
| | `Qwen-3-8B` | SWE-smith Only | 11.0% |
| | `Qwen-3-8B` | ADP Data | **16.6%** |
| *WebArena (Zhou et al., 2024)* | | | |
| AgentLab (Drouin et al., 2024) | `Qwen-2.5-7B-Instruct` | Go-Browse Only | 16.0% |
| (de Chezelles et al., 2025) | `Qwen-2.5-7B-Instruct` | ADP Data | **20.1%** |
| *AgentBench OS (Liu et al., 2024b)* | | | |
| OpenHands CodeActAgent | `Qwen-3-8B` | AgentInstruct Only | 21.5% |
| (Wang et al., 2025) | `Qwen-3-8B` | ADP Data | **25.7%** |
| *GAIA (Mialon et al., 2023)* | | | |
| OpenHands CodeActAgent | `Qwen-2.5-7B-Instruct` | AgentInstruct Only | 0.6% |
| (Wang et al., 2025) | `Qwen-2.5-7B-Instruct` | ADP Data | **9.1%** |

We study whether *data diversity* helps agents generalize across tasks. Holding the agent setup and evaluation fixed, we compare training with different data mixtures: (i) *Base* (no tuning), (ii) *Task-specific only* fine-tuning (e.g., *SWE-smith Only*, etc.), and (iii) *ADP Data* (as detailed in § 5), a mixed, cross-domain corpus. As shown in Table 6, **ADP consistently outperforms task-specific tuning on the *target* task and, critically, avoids the negative transfer that single-domain tuning often induces on *other* tasks** (Mueller et al., 2024; Kotha et al., 2024; Li et al., 2024).

Concretely, on *SWE-Bench*, ADP trained `Qwen-2.5-7B-Instruct` achieves 10.4%, versus 1.0% with *SWE-smith Only*; for `Qwen-3-8B` (Yang et al., 2025a), ADP reaches **16.6%** versus 0.2% with *CodeActInstruct + Code-Feedback* and 11.0% with *SWE-smith Only*. On *WebArena*, ADP trained `Qwen-2.5-7B-Instruct` attains **20.1%** versus 16.0% with *Go-Browse Only*. On *AgentBench OS*, ADP lifts `Qwen-3-8B` to **25.7%** versus 21.5% with *AgentInstruct Only*. On *GAIA*,

*AgentInstruct Only* results in 0.6% accuracy, while ADP improves it to **9.1%**. Overall, mixed ADP tuning yields stronger in-domain accuracy and cross-task generalization than single-domain tuning.

### 6.3 ADP EASES ADAPTATION TO NEW AGENT HARNESSES

Table 7 demonstrates the lines of code (LOC)[3] the authors and community contributors used to convert 13 datasets from distinct sources to the ADP schema. A single *Raw→ADP* converter per dataset performs the same normalization work (schema mapping, tool/action alignment, conversation formatting) that a traditional *Raw→SFT* converter would do for a specific agent harness. Therefore, LOC statistics in Table 7 are a reasonable proxy for the per-agent harness effort *without* ADP.

**Without ADP.** Using this proxy, the cost of converting $D$-many datasets to $A$-many harnesses *without* ADP is $\text{Cost}_{\text{no-ADP}}(A, D) \approx A \cdot \sum_{i=0}^{D} \text{LOC}_{i,\text{Raw}\to\text{ADP}}$. Thus the total conversion cost across the community is ***quadratic*** ($O(D \times A)$ effort), as depicted in Figure 2. In our data, $\sum_{i=0}^{D} \text{LOC}_{i,\text{Raw}\to\text{ADP}} = 4892$ LOC across 13 datasets, so for example for $A = 100$ harnesses the total cost is $\text{Cost}_{\text{no-ADP}} \approx 100 \times 4892 = \mathbf{489,200}$ LOC.

Table 7: LOC for converting datasets to ADP.

| Dataset | Total LOC |
|---|---|
| AgentInstruct | ~1500 |
| Code-Feedback | 134 |
| CodeActInstruct | 269 |
| Go-Browse | 335 |
| Mind2Web | 476 |
| Nebius SWE-Agent Trajectories | 260 |
| NNetNav (live+wa) | 290 |
| openhands-feedback | 879 |
| Orca AgentInstruct | 155 |
| SWE-Gym | 221 |
| SWE-smith | 228 |
| Synatra | 145 |
| **Total** | **4892** |

Table 8: LOC for ADP→SFT converters.

| Agent Harness | Total LOC |
|---|---|
| OpenHands CodeActAgent | ~150 |
| SWE-Agent | ~50 |
| AgentLab | ~30 |
| **Average** | **~77** |

**With ADP.** The total cost becomes $\text{Cost}_{\text{ADP}}(A, D) \approx \sum_{i=0}^{D} LOC_{i,\text{Raw}\to\text{ADP}} + \sum_{j=0}^{A} \text{LOC}_{\text{ADP}\to\text{SFT},j}$ with ADP. Thus, as shown in Figure 2, the total conversion cost across the community now becomes ***linear*** with ADP ($O(D + A)$ effort). Table 8 demonstrates that converting ADP standardized data to agent harness format takes an average of 77 LOC. Across the 13 we used, $\text{Cost}_{\text{ADP}}(A, D) \approx 4892 + 77 \times 100 = \mathbf{12,592}$ for $A = 100$, greatly less than the no-ADP setting. Additionally, adding a new harness only require writing one script converting ADP standardized data to SFT, greatly easing adaptation to new agent harnesses. Hence, **ADP substantially reduces the community's collective effort required to develop scalable, reproducible agents.**

## 7 CONCLUSION AND FUTURE WORK

ADP provides a practical, lightweight "interlingua" that unifies heterogeneous datasets into a single schema consumable by many agent harnesses, turning today's fragmented data landscape into a scalable training pipeline. Looking ahead, we see three immediate directions. **(i) Multimodality:** extending ADP beyond text to images, screen recordings, and other modalities to capture richer agent–environment interactions. **(ii) Standardized evaluation:** applying the same standardized "protocol" idea to evaluation and environment settings so that datasets, agents, and evaluations compose cleanly. **(iii) Community growth and data quality:** continuing open-source releases, stronger automated validation or even automated dataset conversion, to sustain scale while preserving quality. We believe that, by lowering integration costs and enabling systematic and scalable training and analysis across sources, ADP can catalyze the next wave of agent-training research and practice.

### REPRODUCIBILITY STATEMENT.

We provide clear pointers to enable independent reproduction of all results. We describe the ADP schema and conversion pipeline (§ 3), allowing others to regenerate the training corpus from raw sources. We list the datasets and their characteristics in § 2.1. The exact training and evaluation

---

[3]All LOC exclude prompt text (e.g., system prompts); only converter code is counted.

setup-including base models, agent harnesses, our SFT pipeline, the evaluation benchmarks and protocol-is specified in § 5. Finally, we release all code and data open source, including the ADP schemas, converters, and scripts referenced above.

## ACKNOWLEDGEMENT

This work was supported in part by a grant from Fujitsu. The training is supported by the CMU FLAME Center. The authors thank Professor Daniel Fried for his insightful feedback. The authors thank NeuLab members and participants of the LTI ICLR Paper Clinic for helpful suggestions.

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

## A USE OF LLMS

We used LLMs to aid and polish writing for style and presentation.

Specifically, LLMs were employed to:

- polish wording, tighten paragraphs, and improve clarity/flow;
- improve latex presentation (e.g., table/figure captions)

## B   ADP EXAMPLE CONVERSION

The conversion pipeline: Raw $\rightarrow$ ADP $\rightarrow$ SFT enables scalable training across diverse agent architectures while maintaining data provenance and semantic structure.

This section demonstrates an example conversion from raw agent interaction data of the Code Feedback dataset (Zheng et al., 2024a) to the standardized ADP format. The transformation process extracts structured observations and actions from a raw conversation between the user and the agent.

### B.1   RAW FORMAT EXAMPLE

The raw format typically contains conversational messages with roles and content:

Listing 1: Raw Format Example from Code Feedback

```
1  [
2    {
3      "id": 13461,
4      "messages": [
5        {
6          "role": "user",
7          "content": "Write a Python script to calculate statistical
                  measures..."
8        },
9        {
10          "role": "assistant",
11          "content": "You're correct. Let me optimize the code...\n\
                  n```python\nimport math\n\ndef calculate_statistics(x,
                  y):\n    # Implementation details...\n    return
                  r_squared, correlation_coefficient, adjusted_r_squared
                  \n```"
12        },
13        {
14          "role": "user",
15          "content": "Execution result: \nR-squared: 0.6\
                  nCorrelation: 3.87\nAdjusted R-squared: 0.47"
16        }
17      ]
18    }
19  ]
```

### B.2   STANDARDIZED ADP FORMAT EXAMPLE

The standardized format structures the same interaction into typed observations and actions:

Listing 2: Standardized ADP Format Example

```
1  [
2    {
3      "id": "13461",
4      "content": [
5        {
6          "class_": "text_observation",
7          "content": "Write a Python script to calculate statistical
                  measures...",
8          "name": null,
9          "source": "user"
10        },
11        {
12          "class_": "code_action",
```

```
13          "language": "python",
14          "content": "import math\n\ndef calculate_statistics(x, y):
                \n    # Implementation details...\n    return
                r_squared, correlation_coefficient, adjusted_r_squared
                ",
15          "description": "You're correct. Let me optimize the code
                by calculating values once and reusing them..."
16        },
17        {
18          "class_": "text_observation",
19          "content": "R-squared: 0.6\nCorrelation: 3.87\nAdjusted R-
                squared: 0.47",
20          "name": null,
21          "source": "environment"
22        },
23        {
24          "class_": "message_action",
25          "content": "<finish> The code executed successfully with
                statistical results...",
26          "description": null
27        }
28      ],
29      "details": {}
30    }
31  ]
```

The conversion process applies several key transformations:

- **Message Classification**: Raw messages are classified into observations and actions based on content analysis.
- **Code Extraction**: Code blocks within assistant messages are extracted as code_action entries.
- **Source Attribution**: User inputs become text_observation with source: "user", execution results with source: "environment".
- **Thought Preservation**: Original function thoughts are preserved in description fields while structured contents are extracted.
- **Action Classes**: Different classes of agent actions (code execution, messaging, tool usage) are explicitly categorized

This standardization enables systematic analysis of agent behaviors, tool usage patterns, and interaction dynamics across different agent implementations and domains.

### B.3 SFT FORMAT EXAMPLE

The standardized ADP format can be further converted to training-ready formats for specific agent frameworks. Here's the example in OpenHands (Wang et al., 2025) SFT format:

Listing 3: OpenHands SFT Format Example

```
1  [
2    {
3      "id": "13461",
4      "conversations": [
5        {
6          "from": "human",
7          "value": "Write a Python script to calculate statistical
                measures..."
8        },
9        {
```

```
10          "from": "gpt",
11          "value": "You're correct. Let me optimize the code...\n\n<
                function=execute_ipython_cell>\n<parameter=code>\
                nimport math\n\ndef calculate_statistics(x, y):\n    #
                 Implementation details...\n    return r_squared,
                correlation_coefficient, adjusted_r_squared\n</
                parameter>\n</function>"
12        },
13        {
14          "from": "human",
15          "value": "EXECUTION RESULT of [execute_ipython_cell]:\nR-
                squared: 0.6\nCorrelation: 3.87\nAdjusted R-squared: 0
                .47"
16        },
17        {
18          "from": "gpt",
19          "value": "<function=finish>\n<parameter=message>\nThe code
                 executed successfully with statistical results...\n</
                parameter>\n</function>"
20        }
21      ],
22      "system": "You are OpenHands agent, a helpful AI assistant..."
23    }
24  ]
```

## C    DATA SAMPLING FOR BALANCED TRAINING

To balance domains and reduce over-represented sources, we resample each dataset with a per-dataset multiplier $w_d$. For dataset $d$ with $n_d$ raw trajectories, we draw $m_d = \lceil w_d\, n_d \rceil$ examples per epoch; if $w_d < 1$ we sample without replacement (downsample), and if $w_d > 1$ we sample with replacement (upsample). This yields an effective mixture proportional to $w_d$ across datasets (and therefore across domains), while keeping the overall epoch size stable.

Table 9: Per-dataset sampling multipliers $w_d$. $w_d < 1$ indicates downsampling; $w_d > 1$ indicates upsampling.

| Dataset | $w_d$ | Direction |
|---|---|---|
| agenttuning_alfworld | 2 | up |
| agenttuning_db | 2 | up |
| agenttuning_kg | 2 | up |
| agenttuning_mind2web | 2 | up |
| agenttuning_os | 2 | up |
| agenttuning_webshop | 2 | up |
| code_feedback | 0.1 | down |
| codeactinstruct | 1 | neutral |
| go-browse-wa | 1 | neutral |
| mind2web | 1 | neutral |
| nebius_SWE-agent-trajectories | 0.2 | down |
| nnetnav-live | 1 | neutral |
| nnetnav-wa | 1 | neutral |
| openhands | 1 | neutral |
| orca_agentinstruct | 0.001 | down |
| swe-gym_openhands_sampled_trajectories | 3 | up |
| swe-smith | 1 | neutral |
| synatra | 0.01 | down |

In practice, we fix a random seed for reproducibility and shuffle the union of sampled examples across datasets each epoch.  This scheme targets a more balanced distribution

across coding, SWE, tool-use, and web-browsing sources by attenuating very large corpora (e.g., `orca_agentinstruct` at $w_d=0.001$) and amplifying under-represented ones (e.g., `swe-gym_openhands_sampled_trajectories` at $w_d=3$).

## C.1 DOMAIN-SPECIFIC DATA FILTERING

Beyond balanced sampling, we apply domain-specific filtering to optimize training effectiveness for each agent framework based on their evaluation focus and capabilities.

**OpenHands and SWE-Agent Training Data.** For OpenHands CodeActAgent and SWE-Agent, which are primarily evaluated on coding and software engineering tasks (SWE-Bench, AgentBench OS, and GAIA), we use only the *non-web* portion of the ADP training corpus. This includes datasets focused on code generation, software engineering, general agent instruction following, and API/-tool usage. Specifically, we exclude web browsing datasets Mind2Web, Go-Browse, NNetNav, and Synatra to avoid potential interference from web-specific interaction patterns that are not applicable to command-line and coding environments. Thus, using the sampling multipliers in Table 9, the total number of training samples used is around 30K. Future experiments could explore different sampling multipliers and examine the effect of each dataset on coding and software engineering tasks.

**AgentLab Training Data.** For AgentLab, which is designed for web browsing tasks and we evaluated exclusively it on WebArena, we use only the *web* portion of the ADP training corpus. This includes datasets focused on web navigation, browser-based task completion, and web-specific agent instruction following (Mind2Web, Go-Browse, NNetNav, and Synatra). We exclude coding and software engineering datasets to ensure the model is optimized for web browsing patterns and UI element interaction without dilution from less compatible domains. Thus, using the sampling multipliers in Table 9, the total number of training samples used is around 20K. Future experiments could explore different sampling multipliers and examine the effect of each dataset on web tasks.

# D PERFORMANCE SCALING

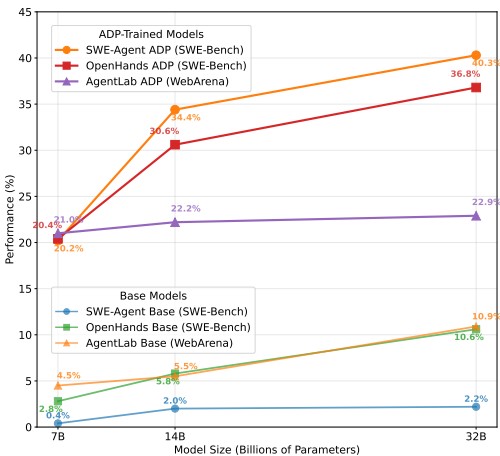
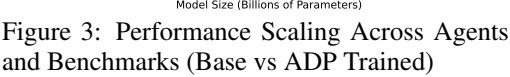

Figure 3: Performance Scaling Across Agents and Benchmarks (Base vs ADP Trained)

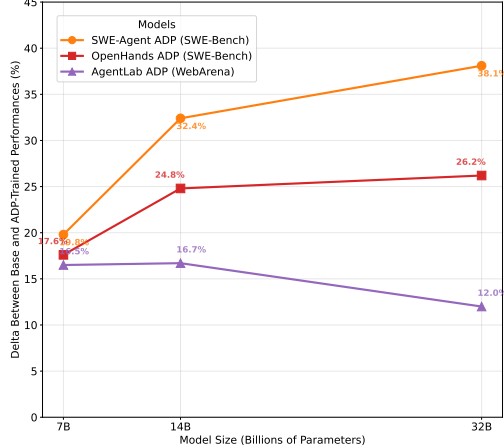

Figure 4: Performance Gains Across Agents and Benchmarks.

Figure 3 and Figure 4 shows the scaling curve of performance and performance gains across agents and benchmarks. Both plots show clear monotonic gains regardless of model size and consistent boosts from ADP training across agents and tasks, with ADP-trained models outperforming their base counterparts at every scale.

# E    ADDITIONAL EXPERIMENTS

## E.1    ADP'S ADVANTAGE PERSIST UNDER EQUAL DATA SCALE

To address the question of fair data scaling, we additionally compare ADP against a single-domain fine-tuning baseline under matched dataset size. Specifically, we train `Qwen-3-8B` on SWE-smith with up-sampling to match the number of training examples used in the ADP mixture, and evaluate both models on SWE-Bench with the OpenHands harness. As shown in Table 10, SWE-smith training yields 11.0% accuracy, whereas ADP training achieves 16.6% under a comparable number of samples. This demonstrates that ADP's benefit does not stem from data volume alone, but from the greater diversity and unified structure of the ADP corpus.

Table 10: Equal-scale comparison of `Qwen-3-8B` trained on SWE-smith vs. ADP, evaluated on SWE-Bench with the OpenHands harness.

| Model | Training Data | Data Scale | Accuracy |
|---|---|---|---|
| `Qwen3-8B` | SWE-smith (up-sampled) | $\approx$ 30K | 11.0% |
| `Qwen-3-8B` | ADP | $\approx$ 30K | **16.6%** |

# F    LICENSE OF USE

This section provides licensing information for all datasets referenced in Table 1 and used in our experiments. We have made every effort to identify and respect the licensing terms of each dataset. Users should verify current licensing terms before using these datasets. Users should also verify the licensing terms of datasets they are adding to ADP.

## F.1    DATASET LICENSES

Table 11: Licensing information for datasets used in ADP

| Dataset | License | Link |
|---|---|---|
| **AgentInstruct** | Apache 2.0 | ZhipuAI/AgentInstruct |
| **Code-Feedback** | Apache 2.0 | m-a-p/Code-Feedback |
| **CodeActInstruct** | Apache 2.0 | xingyaoww/code-act |
| **Go-Browse** | MIT | ApGa/Go-Browse |
| **Mind2Web** | CC BY 4.0 | osunlp/Mind2Web |
| **Nebius SWE Trajectories** | CC BY 4.0 | nebius/SWE-agent-trajectories |
| **NNetNav-live** | Apache 2.0 | stanfordnlp/nnetnav-live |
| **NNetNav-wa** | Apache 2.0 | stanfordnlp/nnetnav-wa |
| **openhands-feedback** | MIT | all-hands/openhands-feedback |
| **Orca Agentinstruct** | CDLA-Permissive-2.0 | microsoft/orca-agentinstruct-1M-v1 |
| **SWE-Gym** | MIT | SWE-Gym/SWE-Gym |
| **SWE-smith** | MIT | SWE-bench/SWE-smith-trajectories |
| **Synatra** | CC BY-SA 4.0 | oottyy/Synatra |

**License Compliance**: We have ensured compliance with licenses of all datasets utilized in this paper. All licenses permit research use.

## F.2    USAGE GUIDELINES

When using the ADP-converted versions of these datasets:

1. **Verify Current Licenses**: Check the original dataset repositories for the most up-to-date licensing terms

2. **Respect Restrictions**: Some datasets have restrictions on commercial use, redistribution, or specific use cases.

3. **Cite Appropriately**: Include citations for both the original datasets and the ADP conversion methodology.

4. **Contact Authors**: For datasets with unclear licensing, contact the original authors for clarification on usage terms.

## F.3 DISCLAIMER

Licenses were collected at the time of dataset integration and may have changed. Users are responsible for verifying current licensing terms and ensuring compliance with all applicable licenses. The ADP project does not assume responsibility for license violations by downstream users.

For questions about specific dataset licenses or usage permissions, please contact the original dataset authors or maintainers directly.

