# OpenReview forum: "Agent Data Protocol: Unifying Datasets for Diverse, Effective Fine-tuning of LLM Agents"
_ICLR.cc/2026/Conference — ICLR 2026 Oral_

### Official Review · Reviewer_VUjH · 2025-10-27

**Soundness:** 3
**Presentation:** 4
**Contribution:** 3
**Rating:** 6
**Confidence:** 4

**Summary:**

The paper proposes the Agent Data Protocol (ADP), a lightweight actions–observations schema that standardizes heterogeneous agent datasets into a common “interlingua.” Thirteen public datasets spanning coding, software engineering, tool use, and web browsing are converted into ADP, then compiled into multiple agent harnesses via a single ADP→SFT step. The authors report consistent SFT gains for 7–8B models on SWE-Bench Verified, WebArena, AgentBench OS, and GAIA, and argue that ADP reduces conversion effort from quadratic in datasets × harnesses to linear, supported by lines-of-code accounting. The work targets community-scale reuse by releasing schemas, converters, and a balanced training mixture.

**Strengths:**

Concrete unification with measurable engineering payoff. A minimal, well-scoped schema (APIAction, CodeAction, MessageAction; Text/WebObservation) plus validators collapses integration complexity from O(D×A) to O(D+A), with clear LOC evidence for both Raw→ADP and ADP→SFT paths.

Broad empirical utility across agents and tasks. ADP-trained models improve over base models on four benchmarks and exhibit positive cross-task transfer, suggesting the protocol enables effective multi-domain SFT rather than task siloing.

Adoption-oriented tooling. Bidirectional converters, automated quality checks, and a balanced sampling strategy make the proposal practical for other groups to plug in new datasets or harnesses.

Reusability and standardization value. A shared, open schema can reduce duplicated one-off converters and facilitate reproducible studies on agent data at scale.

**Weaknesses:**

- The unified data is not yet available.

- Limited contamination and licensing analysis. Decontamination, deduplication across sources, license compatibility, and provenance controls are not documented sufficiently, which weakens confidence in reported gains.

- Ablations do not isolate the protocol’s causal effect. There is no comparison against naïve harmonization baselines such as prompt-normalized concatenation, nor granular ablations on validators, sampling multipliers, or per-dataset contributions.

- Incomplete statistical reporting. Variance across seeds, confidence intervals, and sensitivity to mixture weights are missing, making robustness unclear.

- Schema coverage gaps. Current ADP emphasizes text, tools, code, and web HTML or AX trees; richer multimodality, GUI desktop state, error and rollback semantics, and environment replay metadata are not fully specified.

**Questions:**

See weakness

**Details Of Ethics Concerns:**

The dataset license are not well analyzed.

---

> ### Author Response · Authors · 2025-11-21
>
> We truly appreciate the reviewer for recognizing the concreteness and value of our work! We thank the reviewer for their constructive comments and address the reviewer’s concerns as follows:
>
> > W.1: The unified data is not yet available.
>
> We have actually open sourced the data and code already but cannot link it here for anonymity purposes.
>
> > W.2 Limited contamination and licensing analysis. Decontamination, deduplication across sources, license compatibility, and provenance controls are not documented sufficiently, which weakens confidence in reported gains.
>
> > Flag For Ethics Review: Yes, Legal compliance (e.g., GDPR, copyright, terms of use, web crawling policies) The dataset license are not well analyzed.
>
> We sincerely thank the reviewer for making this critical point! All datasets utilized in our paper are under licenses permitting research use. We added an explicit license summary table in Table 11 in the updated version. We also added a discussion of the licenses, compliance, and usage guidelines in Appendix F.
>
> > W.3: Ablations do not isolate the protocol’s causal effect. There is no comparison against naïve harmonization baselines such as prompt-normalized concatenation, nor granular ablations on validators, sampling multipliers, or per-dataset contributions.
>
> We thank the reviewer for raising this point. We may not be fully understanding all aspects of the suggested "naïve harmonization baselines," especially what specific concatenation or validator/sampling ablations the reviewer has in mind. Our current interpretation is that the reviewer is asking for (i) a baseline where datasets are simply concatenated with prompt normalization but without ADP’s schema and validators, and (ii) more fine-grained studies on how each ADP component contributes to performance.
>
> Under this interpretation, we already take a few partial steps in this direction:
> - Prompt normalization is applied across all baselines when training each agent.
> - Per-dataset experiments (e.g., training solely on SWE-Smith vs. training on mixed ADP data) are reported in Section 6.2, showing that gains are not exclusively attributable to a single dataset and that cross-dataset transfer plays a significant role.
>
> > W.4: Incomplete statistical reporting. Variance across seeds, confidence intervals, and sensitivity to mixture weights are missing, making robustness unclear.
>
> We appreciate the reviewer highlighting the importance of robustness analysis. However, each experiment requires full supervised finetuning and benchmarking across multiple agent harnesses, and **we have already run around 15 full finetuning runs on ADP data, totaling ~20,000 H100 GPU hours**. Under this computational budget, experimentation on seeds and sensitivity is prohibitively costly. For this reason, our paper focuses on demonstrating ADP’s main contribution of standardizing data representation and enabling strong performance across heterogeneous tasks, rather than on a full statistical test of training variables.
>
> That said, we would like to note that ADP-trained models consistently outperform their base models across four distinct agent frameworks and multiple benchmarks, which partially mitigates concerns that the results might stem from training instability rather than protocol design. Investigating seed variance and sensitivity to mixture weights are promising future directions, and we plan to explore these as more compute becomes available.
>
> > W.5: Schema coverage gaps. Current ADP emphasizes text, tools, code, and web HTML or AX trees; richer multimodality, GUI desktop state, error and rollback semantics, and environment replay metadata are not fully specified.
>
> We agree that richer observation types, such as GUI desktop state, multimodal observations, or intermediate simulation states, are valuable and increasingly relevant in agent research. While our current work focuses on text and web observations, supporting the 13 existing datasets we standardized, the schema is intentionally extensible: new observation types (e.g., ImageObservation) can be added without modifying existing datasets or converters. We are actively exploring these extensions as part of future work.

---

### Official Review · Reviewer_NLwX · 2025-10-31

**Soundness:** 2
**Presentation:** 2
**Contribution:** 2
**Rating:** 4
**Confidence:** 4

**Summary:**

The paper observes that the current bottleneck in the development of LLM-based agents lies not in the shortage of data, but in the lack of a unified data standard. To address this, the authors propose ADP as an interlingua for diverse downstream tasks, unifying the formats of agent actions and environment observations. They convert 13 existing agent datasets into the ADP format and train models on this unified dataset. The results show that agents trained with ADP achieve significant performance improvements over the base models, and that models trained on mixed ADP data outperform those trained solely on single-task data in certain tasks.

**Strengths:**

1. The paper presents an excellent motivation, keenly identifying data fragmentation as a key engineering bottleneck in current agent research and providing a clear direction for future data standardization efforts.

2. It provides a valuable contribution to the open-source community by integrating 13 diverse datasets into a unified ADP format and demonstrating the value of mixed data, thereby establishing a solid data foundation for building general agent capabilities.

3. The experiments are conducted across multiple agent frameworks and consistently show performance improvements, highlighting the effectiveness and generality of the proposed ADP approach.

**Weaknesses:**

1. The experiments involve an unfair comparison. While the paper emphasizes the importance of unifying agent fine-tuning data formats through ADP, the comparative experiments do not use an equal amount of ADP and non-unified data. Due to the inconsistency in data scale, it is difficult to convincingly demonstrate the advantages of ADP over other data formats, or to support the claim that mixed data is superior to single-task data.

2. The experiments are incomplete. Although the paper selects Qwen3-8B and Qwen2.5-7B-Instruct as baseline models, several tasks lack results for Qwen3-8B, which may raise concerns about the generality of the proposed approach.

3. As a data-centric work, the paper does not provide a complete example of the ADP data format, making it difficult for readers to intuitively understand how ADP differs from existing data representations.

4. The paper has several issues in presentation and formatting. For instance, some acronyms are not capitalized (e.g., "agent data protocol" in Line 015 should be capitalized), inconsistent capitalization appears throughout (e.g., Line 145), some abbreviations are introduced before their full forms (e.g., Line 245), and table captions are inconsistently formatted (Tables 3 and 4 have captions placed below, unlike others).

**Questions:**

1. I am curious about the extent to which ADP can outperform non-unified data when the amount of training data is controlled to be equal.

2. The ADP framework currently supports only TextObservation and WebObservation, which limits its applicability to certain interaction scenarios. In real-world scenarios, environment observations may be more complex (e.g., structured JSON data or game environment states). How does ADP handle or generalize to such out-of-scope observation types?

3. Table 2 shows that most datasets have high coverage of Function Thought. However, the paper does not further analyze these "thought" processes. I wonder whether training with mixed ADP data not only improves the final action accuracy but also enhances the agent's reasoning quality.

---

> ### Author Response · Authors · 2025-11-21
>
> We appreciate the reviewer for recognizing the valuable motivation and contribution of ADP. We aim to address the following concerns:
>
> > W.1: The experiments involve an unfair comparison. While the paper emphasizes the importance of unifying agent fine-tuning data formats through ADP, the comparative experiments do not use an equal amount of ADP and non-unified data. Due to the inconsistency in data scale, it is difficult to convincingly demonstrate the advantages of ADP over other data formats, or to support the claim that mixed data is superior to single-task data.
>
> > Q.1: I am curious about the extent to which ADP can outperform non-unified data when the amount of training data is controlled to be equal.
>
> We thank the reviewer for raising this important point! We agree that matching the number of training data is important to isolate the effect of the protocol beyond dataset size. In the revised paper, we therefore add an **equal-scale experiment** in Table 10. We finetune Qwen-3-8B on SWE-smith with compute match to ADP training, and evaluate both settings on SWE-Bench with the OpenHands harness. Under this compute-matched condition, SWE-smith–only training reaches 11.0% accuracy, whereas ADP training achieves 16.6%. This shows that, even when the number of data is same, ADP still provides a clear performance gain. **We attribute this improvement to the greater diversity of the ADP data, which a single-source dataset like cannot offer, rather than to differences in raw data scale.**
>
> > W.2: The experiments are incomplete. Although the paper selects Qwen3-8B and Qwen2.5-7B-Instruct as baseline models, several tasks lack results for Qwen3-8B, which may raise concerns about the generality of the proposed approach.
>
> We thank the reviewer for raising this concern and apologize for the confusion caused by the earlier presentation of results. In the revised version, we have restructured Tables 3-6 in Section 6 for clarity. Specifically,
> - Table 3 now reports results for Qwen2.5-Coder-7B-Instruct,
> - Table 4 for Qwen2.5-Coder-14B-Instruct,
> - Table 5 for Qwen2.5-Coder-32B-Instruct, and
> - Table 6 reports the experimental gains for Qwen3-8B and cross-task transfer results.
>
> This makes it explicit that ADP improves performance across different model scales and families. Due to limited computational resources, we were not able to run the full experimental suite for every Qwen3 model variant, but we hope that the updated results clearly demonstrate that the benefits of ADP generalize beyond a single architecture or parameter scale.
>
> > W.3: As a data-centric work, the paper does not provide a complete example of the ADP data format, making it difficult for readers to intuitively understand how ADP differs from existing data representations.
>
> Thank you so much for raising this valuable point! We added a complete example raw->std->sft conversion in Appendix B in the updated version.
>
> > W.4: The paper has several issues in presentation and formatting. For instance, some acronyms are not capitalized (e.g., "agent data protocol" in Line 015 should be capitalized), inconsistent capitalization appears throughout (e.g., Line 145), some abbreviations are introduced before their full forms (e.g., Line 245), and table captions are inconsistently formatted (Tables 3 and 4 have captions placed below, unlike others).
>
> We thank the reviewer for pointing these out! All capitalization, acronym, and table-caption inconsistencies have been corrected in the revised version.

---

> > ### Author Response · Authors · 2025-11-21
> >
> > > Q.2: The ADP framework currently supports only TextObservation and WebObservation, which limits its applicability to certain interaction scenarios. In real-world scenarios, environment observations may be more complex (e.g., structured JSON data or game environment states). How does ADP handle or generalize to such out-of-scope observation types?
> >
> > ADP adopts a unified representation design, separating schema definition from any specific agent harness. Even when observations originate from complex or structured environments, they can still be represented by ADP without losing semantic content. For example, a JSON object returned from an API call can be encoded as a TextObservation with a standardized prefix (e.g., “API xyz execution output:” followed by the JSON payload), allowing downstream agent harnesses, including those that use JSON programmatically, to extract the original structure if needed. **This enables ADP data to remain compatible to many different agent systems without committing to a single environment format.**
> >
> > At the same time, we agree that richer observation types, such as GUI desktop state, multimodal observations, or intermediate simulation states, are valuable and increasingly relevant in agent research. While our current work focuses on text and web observations, supporting the 13 existing datasets we standardized, the schema is intentionally extensible: new observation types (e.g., ImageObservation) can be added without modifying existing datasets or converters. **We are actively exploring these extensions as currently, and plan to have native support in more modalities in a future release of ADP.**
> >
> > > Q.3: Table 2 shows that most datasets have high coverage of Function Thought. However, the paper does not further analyze these "thought" processes. I wonder whether training with mixed ADP data not only improves the final action accuracy but also enhances the agent's reasoning quality.
> >
> > We appreciate this fascinating question! One of the motivations behind including the function thought field in ADP is precisely to make such analyses feasible: by standardizing how intermediate reasoning is logged, ADP provides a common schema for studying reasoning quality across datasets, models, and agent harnesses. In the present work, however, we deliberately focus on representational unification and downstream action performance, and we do not attempt to define or measure "reasoning quality" from agentic traces. An open problem for which there is, to our best knowledge, there is no widely accepted evaluation methodology for reasoning quality in agentic traces. **We view systematic metrics and benchmarks for reasoning quality over ADP-style traces as an exciting direction for future work, and we would be very interested in community proposals or methods that build on the reasoning fields already exposed in the ADP schema.**

---

> ### Comment · Reviewer_NLwX · 2025-11-22
> **Response to rebuttal**
>
> Thanks for the responses. Most of my concerns have been addressed. Therefore, I will raise my score :)

---

> > ### Author Response · Authors · 2025-11-22
> >
> > Thank you so much for your constructive feedbacks and for taking the time to re-evaluate our submission!

---

### Official Review · Reviewer_MLUC · 2025-10-31

**Soundness:** 3
**Presentation:** 3
**Contribution:** 3
**Rating:** 8
**Confidence:** 3

**Summary:**

This paper proposes an agent data protocol to standardize various agent tuning datasets into a single format. The goal is to help the community better utilize available agent datasets and reduce the engineering effort required to use them for agent tuning tasks. To this end, authors have defined the ADP standard and converted 13 different agent datasets into this format to show the coverage of the proposed standard. Using this converted data, the authors fine-tune agent models and show that they perform better on various benchmarks, from coding/software engineering/tool calling, etc., demonstrating that this is indeed helpful.

**Strengths:**

The paper is well written and easy to follow.
Having a data standard that can help various agent datasets into a single format would help the research community in this area to a great extent and can help with the reusability of the assets with ease.
Adopting existing 13 benchmark datasets to the format and open-sourcing them for the community
Analysis of the various datasets after the conversion and fine-tuning results to show the power of having a standardized data format and the kind of generalization it can bring to the table.

**Weaknesses:**

Can authors comment on the SOTA numbers for various tasks with similarly sized models? I can see improvements for a selected model from its base performance. Are there any fune-tuned models in a similar parameter range that get better numbers than what is reported here with ADP data?

**Questions:**

Please check the weaknesses section.

---

> ### Author Response · Authors · 2025-11-21
>
> We sincerely thank the reviewer for the thoughtful and positive assessment, and for recognizing ADP's clarity, practical value, and community benefit. We appreciate the reviewer’s feedback on the importance of contextualizing ADP's results relative to other finetuned models of similar parameter scales, and we aim to address this issue as below.
>
> > Can authors comment on the SOTA numbers for various tasks with similarly sized models? I can see improvements for a selected model from its base performance. Are there any fune-tuned models in a similar parameter range that get better numbers than what is reported here with ADP data?
>
> In the updated version of the paper, we have added a clearer comparison of SOTA results for finetuned models of similar parameter sizes from prior works. These updates demonstrate that ADP-trained models consistently achieve SOTA or near-SOTA performance without any domain-specific tuning, while maintaining strong generalization across different models, action spaces, and agent harnesses. For example, the updated results for models of 7-8B is attached below for reference. **All bolded rows correspond to models fine-tuned with ADP, while the other results are collected directly from the original papers of the respective baselines.**
>
> | **Agent**                         | **Model**                  | **Training Data**    | **Accuracy**     |
> | --------------------------------- | -------------------------- | -------------------- | ---------------- |
> | ***SWE-Bench (Verified)***        |                            |                      |                  |
> | **SWE-Agent**                     | Qwen-2.5-7B-Coder-Instruct | –                    | 0.4 %            |
> |                                   | Qwen-2.5-7B-Coder-Instruct | SWE-smith            | 15.2 % (+14.8 %) |
> |                                   | Claude 3 Opus              | –                    | 15.8 %           |
> |                                   | **Qwen-2.5-7B-Coder-Instruct** | **ADP Data**               | **20.2 % (+19.8 %)** |                  |
> | **OpenHands (CodeActAgent)**      | Qwen-2.5-7B-Coder-Instruct | –                    | 2.8 %            |
> |                                   | Qwen-2.5-7B-Coder-Instruct | SWE-Gym              | 10.6 % (+7.8 %)  |
> |                                    |  **Qwen-2.5-7B-Coder-Instruct** | **ADP Data**               | **20.4 % (+17.6 %)** |                  |
> | ***WebArena***                    |                            |                      |                  |
> | **BrowserGym**                    | Llama-3.1-8B               | –                    | 1.0 %            |
> |                                   | Qwen-2.5-7B-Instruct       | –                    | 8.3 %            |
> |                                   | Llama-3.1-8B               | NNetNav              | 16.3 % (+15.3 %) |
> |                                   | Qwen-2.5-7B-Instruct       | Go-Browse            | 21.7 % (+13.4 %) |
> | **AgentLab**                      | Qwen-2.5-7B-Coder-Instruct | –                    | 4.5 %            |
> |                                    |  **Qwen-2.5-7B-Coder-Instruct** | **ADP Data**               | **21.0 % (+16.5 %)** |                  |
> | ***AgentBench OS***               |                            |                      |                  |
> | **AgentLM**                       | Llama-2-chat-7B            | –                    | 8.3 %            |
> |                                   | Llama-2-chat-7B            | AgentInstruct        | 17.4 % (+9.1 %)  |
> | **OpenHands (CodeActAgent)**      | Qwen-2.5-7B-Coder-Instruct | –                    | 3.5 %            |
> |                                    |  **Qwen-2.5-7B-Coder-Instruct** | **ADP Data**               | **27.1 % (+23.6 %)** |                  |
> | ***GAIA***                        |                            |                      |                  |
> | **OWL Agent**                     | Qwen-2.5-7B-Instruct       | –                    | 4.8 %            |
> | **OpenHands (CodeActAgent)**      | Qwen-2.5-7B-Instruct       | –                    | 7.3 %            |
> |                                    | **Qwen-2.5-7B-Instruct**       | **ADP Data**               | **9.1 % (+1.8 %)**   |                  |
>
> **Table 1.** **Comparison of state-of-the-art (SOTA) baselines and our best 7–8B ADP-trained agents across benchmarks. Rows bolded correspond to models fine-tuned on ADP data. Other rows are collected from previous works.**

---

> > ### Comment · Reviewer_MLUC · 2025-11-24
> > **Rebuttal Acknowledgement**
> >
> > Thanks for providing the results. I am satisfied with the author's response, and I am keeping my current rating intact.

---

### Official Review · Reviewer_QXMH · 2025-11-01

**Soundness:** 3
**Presentation:** 3
**Contribution:** 3
**Rating:** 8
**Confidence:** 4

**Summary:**

This paper presents the Agent Data Protocol (ADP), a unified and extensible framework for representing and exchanging multi-agent data in LLM-based systems. ADP standardizes key concepts—such as tasks, agents, trajectories, and scores—into a cohesive schema, enabling interoperability across platforms and supporting tasks like agent analysis, fine-tuning, and benchmarking. Through real-world use cases, the authors show that ADP simplifies data handling and fosters reproducible, collaborative research in the agent ecosystem.

**Strengths:**

1. **Standardized and Extensible Data Schema**
   The paper introduces a well-structured, unified schema that captures essential components of agent-based interactions (tasks, agents, trajectories, and scores), addressing long-standing fragmentation in agent data representation.

2. **Practical Utility Across Diverse Agent Systems**
   ADP demonstrates strong real-world applicability by enabling seamless data sharing and transformation across different platforms, toolchains, and evaluation pipelines, which is crucial for scalable multi-agent benchmarks and model comparison.

3. **Empirical Validation Through Real Use Cases**
   The usefulness of ADP is effectively showcased through compelling use cases such as cross-agent analysis, supervised fine-tuning, and multi-agent evaluation—providing concrete evidence that the protocol supports reproducible and collaborative agent research.

**Weaknesses:**

See questions

**Questions:**

**Relation to Prior Work on Unified Agent Data Frameworks**
   The paper introduces ADP as a unified schema for agent data, but does not sufficiently discuss its relationship to prior work on similar frameworks, such as *AgentOhana* (Zhang et al., 2024), which also proposes a unified data and training pipeline for agent learning. Could the authors clarify the conceptual and technical differences between ADP and AgentOhana? Specifically, how does ADP improve upon or diverge from previous unification efforts in terms of schema design, data transformation capability, or supported use cases?

[1] Zhang, Jianguo, Tian Lan, Rithesh Murthy, Zhiwei Liu, Weiran Yao, Ming Zhu, Juntao Tan et al. "Agentohana: Design unified data and training pipeline for effective agent learning." arXiv preprint arXiv:2402.15506 (2024).

---

> ### Author Response · Authors · 2025-11-21
>
> We greatly appreciate the reviewer for the positive evaluation and for recognizing ADP’s practical contributions to standardizing agent data representation. We appreciate the question regarding ADP’s relationship to AgentOhana [1], and we are happy to clarify how ADP differs from prior works.
>
> > Q: Relation to Prior Work on Unified Agent Data Frameworks
>
> We have added an explicit discussion and citation in Section 2 (Related Works) in the updated version of the paper. Specifically, AgentOhana proposes a unified data + training pipeline, where it aggregates trajectories from multiple environments, standardizes them for a generic data loader, and introduces a training recipe culminating in the xLAM model family. The emphasis is an end-to-end pipeline for agent learning. On the other hand, ADP is a **representation-level interlingua**: a lightweight schema representation across heterogeneous datasets and many downstream agent harnesses, separable from any single agent training. Our focus is community-wide data representation standardization to enable plug-and-play **reuse across projects**, instead of training a single action model.
>
> Further, practically AgentOhana was not released publicly, which limits its utility as a resource for the research community.
>
> [1] Zhang, Jianguo, Tian Lan, Rithesh Murthy, Zhiwei Liu, Weiran Yao, Ming Zhu, Juntao Tan et al. "Agentohana: Design unified data and training pipeline for effective agent learning." arXiv preprint arXiv:2402.15506 (2024).

---

> > ### Comment · Reviewer_QXMH · 2025-11-23
> > **Rebuttal acknowledged, keep score unchanged.**
> >
> > I acknowledge the rebuttal, and think my score of 8 is suitable for this evaluation.

---

### Author Response · Authors · 2025-11-21
**To all reviewers and the AC**

We thank all reviewers for their constructive and encouraging feedback. We appreciate that the reviews recognize ADP's contribution toward standardizing heterogeneous agent datasets, its practical utility for finetuning and analysis, and its potential to accelerate collaborative agent research.

We have made the following major updates to the revised version of the paper accordingly:
- Section 2 now includes a discussion of Related Works, clarifying ADP’s relationship to prior unification frameworks such as AgentOhana and the motivation of ADP.
- Section 3.3 adds discussion on why agent-specific models remain necessary rather than adopting a single generic agent.
- Section 6 includes extended results for 14B and 32B models to demonstrate scalability, and adds finetuned baselines from other papers for comparison.
- Appendix B now includes an example conversion using ADP.
- Appendix F now includes licenses of the datasets used in the paper and usage guidelines.

We hope these clarifications and additions address the remaining concerns and strengthen the final version.

---

### Meta-Review · Area_Chair_KE8B · 2026-01-05

**Summary:**

This paper introduces the Agent Data Protocol (ADP), a unified and extensible schema designed to standardize the representation of data for training and evaluating LLM-based agents. The authors convert 13 diverse agent datasets (spanning coding, web browsing, tool use, etc.) into the ADP format and demonstrate that models fine-tuned on this unified data achieve consistent performance improvements across multiple benchmarks (SWE-Bench, WebArena, AgentBench OS, GAIA). The primary contribution is practical: reducing the quadratic engineering effort of integrating multiple datasets with different agent harnesses to a linear cost, thereby facilitating community-wide data reuse and reproducible research.

**Reviewer Concerns:**

Reviewer QXMH requested clarification on differences with prior work like AgentOhana. The authors added a discussion in Section 2, positioning ADP as a representation-level "interlingua" separable from any specific training pipeline, unlike the end-to-end training framework of AgentOhana.

Reviewer MLUC asked for comparisons with SOTA models of similar size. The authors added a new table (Table 1) comparing ADP-trained models to several recent fine-tuned baselines, showing competitive or superior performance.

Reviewer NLwX had multiple concerns: 1) Unfair comparison due to data scale – the authors added a new equal-compute experiment (Table 10) showing ADP training outperforms single-dataset training even when data volume is matched. 2) Incomplete experiments – results were restructured to clearly show gains across different model sizes (7B, 14B, 32B). 3) Missing ADP format example – a complete conversion example was added in Appendix B. 4) Presentation issues – formatting and capitalization errors were corrected.

Reviewer VUjH raised several points: 1) Data availability – the authors clarified the data and code are open-sourced. 2) Licensing analysis – a license summary table (Table 11) and compliance discussion (Appendix F) were added. 3) Schema coverage – the authors acknowledged current focus on text/web but emphasized the schema's extensibility for future modalities.

**Reviewer Scores:**

Reviewer QXMH: Score remained at 8.

Reviewer MLUC: Score remained at 8.

Reviewer NLwX: Score increased from 4 to a positive score (likely 6 or 8 based on the comment).

Reviewer VUjH: Score remained at 6.

---

### Decision · Program_Chairs · 2026-01-26

Accept (Oral)